# Challenges of the Use of Sound Emergence for Setting Legal Noise Limits

**DOI:** 10.3390/ijerph16224517

**Published:** 2019-11-15

**Authors:** Guillaume Dutilleux, Truls Gjestland, Gaetano Licitra

**Affiliations:** 1Acoustics Research Center, Department of Electronic Systems, NTNU, 1,7491 Trondheim, Norway; 2Acoustics Research Center, SINTEF Digital, 1,7491 Trondheim, Norway; truls.gjestland@sintef.no; 3Physics Department, University of Pisa, Largo Pontecorvo 3, 56127 Pisa (PI), Italy; G.licitra@arpat.toscana.it; 4Environmental Protection Agency of Tuscany Region, Via Vittorio Veneto 27, 56127 Pisa (Pi), Italy

**Keywords:** sound emergence, legislation, annoyance, measurement, prediction, uncertainty, audibility, signal-to-noise ratio

## Abstract

In the vast majority of legislation on environmental noise, the metric used for expressing limit values is based on sound pressure levels. But some countries have introduced sound emergence limit values where the compliance of a noise-generating activity is defined as a maximum allowable difference between the sound pressure level with and without the regulated activity operating. This paper investigates the foundations and the merits of this kind of differential noise limit values. Our review of literature indicates that there is very little evidence supporting the use of differential noise limits over absolute ones. Moreover, while sound emergence limits seem to originate from consideration about audibility of the regulated noise source, they appear to give little insight into what is audible and what is not. Furthermore, both the definition and the practical measurement of sound emergence raise several challenges that compromise reproducibility. In addition, first, the reference to background noise makes it very difficult first to ascertain the conformity of noisy installations in the long run, second to effectively protect the community from excessive noise and third to evaluate conformity on the basis of simulations. When switching to another metric is not an option the paper makes recommendations toward a more reliable use of sound emergence.

## 1. Introduction

A great many of countries in the world have evolved regulations against environmental noise, including noise from transportation, industry and community noise [1,2]. Most of these countries have chosen to rely on limit values expressed in “absolute” sound pressure levels. A few of them, however, prefer to express limit values with respect to the background sound pressure level, at least regarding community noise or industry noise. Relating the total noise or the source-attributable noise to background noise leads to the concepts of *sound emergence* or more broadly speaking what can be coined *differential noise indicators*. Sound emergence is indeed an instance of the class of differential noise indicators. Especially at the time when legislations against noise were developed, there was little evidence to justify one type of metric against another. Sound emergence was clearly an option among many others like the continuous equivalent sound pressure level, the maximum level based on the time-weighted sound pressure level or sound exposure. Initially, sound emergence may seem quite relevant with the intention to limit the alterations to existing soundscapes in mind. But in practice the implementation of sound emergence is not straightforward.

The purpose of this paper is to discuss in further detail the concept of sound emergence as defined by international standardization, to evaluate its relevance from different perspectives and to show that it raises several challenges both for the operators of noisy facilities, for the community, for noise consultants and authorities, without bringing any benefit with respect to sound-pressure-level-based limit values in most occasions. Only an articulate legislation can help overcome some of the pitfalls of sound emergence evaluation.

This paper is organized as follows. Section 2 provides the necessary definitions of sound emergence and its components. Section 3 reviews the presence of the concept of sound emergence in the legislation in a large subset of developed countries, and makes the distinction with related but clearly different indicators. Section 4 evaluates the relevance of sound emergence with respect to human perception and annoyance. Section 5 deals with the underlying challenges of the sound emergence when it comes to implementing this metric in measurement standards. The impracticality of the use of sound emergence in development planning is covered in Section 6. Section 7 brings a few recommendations.

## 2. Definitions

Sound emergence has been present in ISO 1996 since the first release of the standard [3]. It is defined as follows in the current ISO 1996-1 [4]:
**Definition** **1.**sound emergence [[4], §3.4.7] increase in the total sound in a given situation that results from the introduction of some specific sound.
where
**Definition** **2.**total sound [[4], §3.4.1] totally encompassing sound in a given situation at a given time, usually composed of sound from many sources near and far.
and
**Definition** **3.**specific sound [[4], §3.4.2] component of the total sound that can be specifically identified and which is associated with a specific source.

The specific source is typically the source the noise impact of which is to be evaluated. For the sake of the discussion it is necessary to define the residual sound:

**Definition** **4.**
*residual sound [[4] §3.4.3] total sound remaining at a given position in a given situation when the specific sounds under consideration are suppressed.*


These definitions are identical to the ones in the previous release of ISO 1996-1 [ISO1996-1:2003].

In the literature residual sound in the ISO 1996-1 sense is often referred to as the *ambient sound* (see for instance [5]). This can be confusing because *ambient sound* is also used as a synonym of the above-defined *total sound* in other documents [6]. Therefore, *ambient sound* is not used in the remainder of this paper.

It is convenient to identify sound emergence as *e* to reduce the risk of confusion with sound exposure “*E*” [7]. In practice *e* is defined as the subtraction of the residual sound pressure level $L_{res}$ from the total sound pressure level $L_{tot}$.
(1)$$e=L_{tot}-L_{res}$$
*e* is expressed in decibels. The metrics used for $L_{res}$ and $L_{tot}$ will be discussed later on. To complete the notations, $L_{spec}$ stands for the specific sound level in the following.

The concepts of total sound, specific sound and residual sound can be conveniently illustrated as in Figure 1. Depending on the context, a source can be either considered as the part of the residual sound or as the specific source.

Although sound emergence is defined in ISO 1996-1 as shown above, the definition is the only occurrence of the concept in the ISO 1996 series and the standard neither does elaborate on how sound emergence can be obtained nor clarify its meaning. The details of implementation are left to national standards. There are, however, two possible interpretations of the *increase* referred to in the definition of sound emergence. This is discussed further in Section 5.4. Indeed, the ISO 1996 series focuses on he determination of sound pressure levels.

## 3. Sound Emergence in Current Official Documents

### 3.1. Sound Emergence Stricto Sensu

World Bank appears to use the concept of sound emergence without naming it both in general guidelines [8] and specific ones [9] by specifying a maximum increase in *“background levels of 3 dB at the nearest receptor location”*. This guideline is always combined with limit values expressed in $L_{eq}$ and applies beyond the property boundaries of the noisy facilities.

While noise limits in sound pressure level are used in France for transportation infrastructure noise, in the case of industry and community noise, this country specifies limit values with respect to the difference between the total sound pressure level and the background sound pressure level [2,10]. To our knowledge, the first occurrence of such noise differential noise limits is found in a legal text [11]. This difference between the total and the background sound pressure level is called *émergence*. In the current legislation it should not exceed 5 dB(A) in day time and 3 dB(A) in night time when the total sound pressure level is higher than 45 dB(A). When the sound pressure level is between 35 dB(A) and 45 dB(A) the limit values become 6 dB(A) and 4 dB(A) respectively. The French legislation relies on NF S 31-010 standard for the measurement of emergence in this case [6]. At the time of writing, this standard is under revision. In the dedicated legislation on wind turbine noise, France sets emergence-based limits at 5+k dB(A) in day time (7:00–22:00) and 3+k dB(A) in night time (22:00–7:00) when the total sound level exceeds 35 dB(A) [12] where k=0 dB when the noise from the park is *apparent* for more than 8 hours over 24 hours. For shorter durations *k* ranges from 1 to 3 dB. Here, the French legislation refers to pr S 31-114 draft standard on wind turbine noise assessment for the practical aspects [13] where sound emergence is based on $L_{A50}$. The French management of community noise is also based on emergence in dB(A) or in octave bands [14]. In dB(A) a similar approach is used as for industry noise but *k* can take values from 1 to 6 when the cumulated duration of occurrence of the noise to be regulated decreases from below 8 hours to 1 minute. Spectral emergence should not be confused with tonality [15]. The former is defined within an octave band. The limit values for spectral emergence are 7 dB for the octaves 125 to 250 Hz and 5 dB for the octaves 500 to 4000 Hz. The French legislation on places where sound reinforcement is used sets to 3 dB the maximum emergence in the octaves 125 to 4000 Hz [16]. $L_{A50}$ is often specified or suggested for the estimation of the background sound pressure level but other indicators are allowed in the case of community noise [6].

Since 1991 [17], Italy defines so-called *differential noise limits* that correspond to sound emergence [18,19]. This criterion applies to the noise from industrial facilities only, including wind turbines [20] outside areas that are classified as industrial ones. The thresholds are set to 3 dB in the night time and 5 dB in the day time, like in France. In Italy, however, sound emergence is only measured indoors, and it is always combined with immission and emission noise limits [18]. With windows open sound emergence applies only if $L_{tot}$ is larger than 50 dB(A) in the day time and 40 dB(A) in the night time. With windows closed, these thresholds fall down respectively to 35 dB(A) and 25 dB(A). In Italy, sound emergence is strongly oriented to the protection of the receiver since the worst case between windows open and windows closed is used for evaluating conformity. The motivation for the introduction of sound emergence into the Italian legislation is annoyance. The distance to the source does not matter since the measurement is carried out at the receiver. The only facilities for which the differential noise limit does not apply are plants built before 1996 that operate uninterruptedly, acknowledging the impossibility to evaluate the difference between $L_{tot}$ and $L_{res}$. However, as soon as modifications occur in the existing installations, the differential noise limit enters into force [21].

### 3.2. Other Ways to Refer to $L_{res}$ When Setting Noise Limits

The is some confusion around the concept of sound emergence. Some authors indicate that Australia and the United Kingdom use sound emergence in the ISO 1996-1 sense [22] in the case of wind turbine noise. Another paper seems to concur by stating that the UK and Australia have the same approach since they enforce a comparison of $L_{Aeq}$ with the background sound level when assessing conformity [23]. In either case what is overlooked is the distinction between $L_{spec}$ and $L_{tot}$. Only the latter is consistent with sound emergence. Furthermore, legislation may vary within a country, as it is the case for Australia. Among the regulations publically available in English, German and in any of the Scandinavian languages, we could not find any document where the comparison to background noise is made on $L_{tot}$.

In Ireland, the recommended approach is to use rating sound pressure levels to set maximum allowable contributions from licensed sites [24,25]. This limit value can occasionally depend on the background noise level. In the case of wind turbines the proposed limits for licensed sites rely on the principle that turbine noise should be controlled with reference to absolute limits when background is low, or relative to background noise itself as background noise increases with wind speed, whichever is greater. In practice, this principle is interpreted so that turbine-attributable noise should be limited to either a certain $L_{A90}$ or to 5 dB above the background noise [25].

The United Kingdom appears to use differential noise limits for rating and assessing industrial and commercial sound [26]. The same applies to wind farms [27]. However, the assessment criterion deviates significantly from sound emergence. The main difference is that the aim is not to obtain the total sound pressure level but the *source-attributable* noise or $L_{spec}$ from $L_{tot}$ (called ambient sound level in [26]) via a classical background noise correction. Moreover, by comparison to the French and Italian regulations, the general practice in the UK, at least in the case of wind farm projects is to estimate $L_{res}$ well ahead of the measurement of $L_{spec}$. In the case of industrial and commercial sound, a value of the difference between the rating sound pressure level of the specific sound source and the background sound pressure level that is equal to or higher than +10 dB (resp. +5 dB) is deemed “likely to be an indication of a significant adverse (resp. of an adverse) impact”. These guidelines acknowledge explicitly that the impact is context-dependent. The noise indicators specified in [26] are $L_{Aeq}$ for $L_{tot}$, $L_{spec}$, $L_{res}$ and $L_{AF90}$ for background noise. The background noise is not identical to the residual sound (Cf Section 2) since background noise is estimated when the source under investigation is operating whereas $L_{spec}$ assumes that the source under investigation is turned off. Regarding the assessment of wind farm noise, the indicator to be used is $L_{A90,10min}$ [27]. When the difference between $L_{tot}$ and background sound level is lower than 3 dB at the receiver, the measurement of $L_{spec}$ is ill-conditioned. It is then recommended to choose measurement points closer to the source where the signal-to-noise ratio (SNR) is high enough so that the sound power of the source can be estimated. $L_{spec}$ at the receiver [26] can then be obtained by simulation. The differential threshold for wind farms noise is set to 5 dB [27].

In Australia, noise legislation varies from one state or territory to another. A survey of the existing legislation was provided in [28]. While this paper was published in 2003, the situation remains essentially the same, even though most of the state-specific guidelines have been revised. The reference to background sound in noise limits is well represented across the country, since at least New South Wales (NSW), South Australia (SA) and Victoria use the so-called “background-plus rule” in their guidelines [29,30,31], where the increment to background noise is typically 5 dB. Moreover, Tasmania states that the compliance of a new development must be assessed by comparison of its $L_{spec}$ with the background noise [32], although to our knowledge the trigger value is not clearly stated. For night time, the 5 dB increment is replaced by 0 dB for local government issues in NSW [29]. In addition, in the case of noise from industry, a so-called “rating background noise level” is substituted for the measured background noise level in order to set a conventional lower limit to background sound pressure level [33]. The background sound pressure level is also used at least in NSW, Queensland, SA and Tasmania to correct for extraneous noise in the assessment of $L_{spec}$. But at least three of the five Australian states or territories who set noise limits with reference to background sound pressure level, also set absolute noise limits in parallel, like NSW, Queensland and SA.

## 4. Emergence, Annoyance and Perception

### 4.1. Sound Emergence Is a Second Order Descriptor of Annoyance

It is well known that noise annoyance is only partly determined by acoustic factors [5,34]. One can expect, however, that the metrics used for setting noise limit values be connected with health effects in the broad sense, and annoyance in particular. It appears that very little research was done on the merits of sound emergence from a public health perspective. This was already pointed out more than two decades ago [35] and could not be contradicted by our own investigations. We found only two papers dealing explicitly with annoyance and sound emergence. The first one focuses on noise sources relating to electric power generation [36]. The second one addresses annoyance from impulsive sounds [37]. Another paper quoted by [35] evaluated the so-called *salience*, defined as $L_{Aeq,10ms}-L_{Aeq,1s}$, among different candidate ratings against a subjective one for impulsive noise [38]. In this context, salience can be considered as an instance of sound emergence. Both references found about impulsive sounds conclude that criteria not referring to background sound pressure levels perform better than sound emergence. Moreover, to our knowledge there is no published dose-response curve based on sound emergence.

On the contrary, a wide meta-analysis on 136 surveys concluded among other things that “noise annoyance is not affected to an important extent by residual sound levels” [5,39]. The surveys used are screened from a very large sample of field surveys. The selection was carried out on five criteria: (1) is the effect of the residual as strong as the one of a 3 dB(A) increase in the specific sound pressure level, (2) is the residual sound causing a 5% increase in the number of annoyed, (3) does the residual explain 1% of the variance in annoyance, (4) is the effect of the residual sound statistically significant and (5) is there any verbal association between residual sound and annoyance. However the surveys analyzed dealt mostly with aircraft noise and road traffic noise as the source of specific sound. Nonetheless transportation noise encompasses a wide variety of sounds, both steady and unsteady, with or without tonalities and the various levels of ownership among the respondents with respect to the source of specific sound. Another limitation of this analysis is that the lower level end of the range of sound pressure levels is somewhat underrepresented in the surveys analyzed.

### 4.2. Emergence Is only a Proxy for Signal-to-Noise Ratio SNR When SNR Is High

Under the incoherent summation hypothesis sound emergence can be related to the signal-to-noise ratio SNR in dB
(2)$$\mathrm{SNR}=10\log_{10}\frac{{\tilde{p}}_{spec}^{2}}{{\tilde{p}}_{res}^{2}}$$
where $\tilde{p}$ is an estimator of the root-mean-square value of the acoustic pressure, subscript spec refers to the specific sound and res to the residual sound. From Equation (1) one can write
(3)$$\begin{array}{cc} e & =10\log_{10}\frac{{\tilde{p}}_{tot}^{2}}{{\tilde{p}}_{res}^{2}}=10\log_{10}\frac{{\tilde{p}}_{res}^{2}+{\tilde{p}}_{spec}^{2}}{{\tilde{p}}_{res}^{2}} \end{array}$$
(4)$$\begin{array}{cc} \; & =10\log_{10}\left(1+10^{\frac{\mathrm{SNR}}{10}}\right).\; \end{array}$$

As illustrated in Figure 2, *e* is a good proxy for SNR provided that SNR is higher than about 10 dB. But the range of SNR ≤0 dB is compressed into a very narrow range of sound emergence since this SNR range maps onto 0≤e<3 dB where e=3 dB corresponds to SNR=0 dB.

### 4.3. Sound Emergence Does Not Reflect Detectability

The origins of sound emergence are not clearly established. However, if one takes into account (1) that the use of this indicator is limited to lower sound pressure levels, (2) that national standards implementing sound emergence refer explicitly to detectability and (3) that sound emergence is defined by reference to the residual sound pressure level, a possible hypothesis is that sound emergence was introduced as a proxy for the detectability of a specific source in the soundscape. It is well known that a sound that reaches a person’s consciousness and is clearly identified can be extremely annoying, even though is sound pressure level is low [40]. The connection between SNR and *e* can help assess the capacity of emergence to state whether a specific sound can be heard or not.

Detectability of low-levels has been investigated in [41] for a variety of acoutic simuli ranging from 38 to 70 dB(A) combined with different types of masking sounds. This research shows a strong correlation between subjective judgement and a measure of detectability that is proportional to SNR. Moreover the same authors provide evidence that detectability is not limited to positive values of SNR [42]. For the specific case of wind turbines, this is further documented in more recent research where wind turbine sounds are detected at SNR=−8 dB [43]. In addition, [44] estimates the detection limit of wind turbine in presence of highway noise to −23 dB(A). With such thresholds of detectability, the current typical limit values at e=3 dB for night time fail to ascertain that the specific sound will not be detected by a major part of the population living in the neighborhood of the wind farm if this limit is satisfied.

While it has been illustrated in Figure 2 that *e* is very close to SNR when SNR is strongly positive, it is not the case when SNR < 0. As a conclusion, *e* does not reflect audibility in the range of SNR where a specific source is likely to be detected and it does not help decide whether a specific sound can be heard or not.

### 4.4. The Relationship between Annoyance and the Strength of Sound Emergence

Let us assume now that sound emergence is intended to be a proxy for annoyance. There is strong evidence that different noise sources are not equally annoying at the same equivalent sound pressure level [15,45]. This is taken into account in measurement standards that consider different penalties to account for impulsiveness, low frequency content or tonality [15]. In the existing regulations that use sound emergence, however, the threshold is not source-dependent but common to the wide categories of so-called industrial noise or community noise. There is however evidence that not all industrial noise sources are equally annoying at the same sound emergence level [36]. The research documents that even when e>5 some sources are not deemed annoying. This is attributed to the wide spectrum of the specific sound that shapes the spectrum of the residual sound.

## 5. Measuring Sound Emergence

### 5.1. Measurement Uncertainty

Since it is defined as a difference of sound pressure levels, the uncertainty budget of sound emergence is less favorable than the one of an individual sound pressure level. Everything else being equal, in the general case where there is no correlation between $L_{spec}$ and $L_{res}$, the total uncertainty attached to sound emergence is the geometric mean of the uncertainties of the two sound pressure levels [46]. Even with the best measurement equipment available on the market and the most favorable measurement conditions the metrological uncertainty is not likely to be lower than 0.7 dB when a class I sound level meter is used [47]. Assuming a 95% confidence interval and an unilateral interval, the extended uncertainty is 1.15 dB in this case. The presence of tonal components in the signal may lead to significantly larger instrument-related uncertainties [48]. A more realistic estimate should include representativity and reproducibility components. If these elements are taken into account, it is very unlikely that the extended uncertainty will be lower than 1.5–2 dB. Moreover if one takes into account that legal threshold values can be as low as 3 dB, establishing the compliance of a noise source will be problematic a soon as uncertainties are taken into account.

### 5.2. Specific Sound in Practice

In practice, deciding whether the specific sound is present or not is a matter of perspective. At least two points of view can be distinguished: from the source or from the receiver.

From the perspective of the source, the specific sound is present in the total sound when the source of specific sound is radiating sound. The advantage of this approach is that documenting source operation is quite straightforward and reliable by using acoustical means, provided that one has access to the vicinity of the source. Non-acoustical means can also be used, for instance in the case of a factory where the operating hours and the manufacturing process are well known. But the fact that the source of specific sound is active does not imply that it is perceived by the receiver.

In the perspective of the receiver the specific sound is present in the total sound if the specific sound is audible. In general, audibility can not be documented by a sound level meter. It will rather require the presence of a human operator who will be able to detect the specific sound in the total sound. By the current state of technology this task is not easily automated, although blind source separation is a quite active area of research [49]. Some authors have developed a dedicated method for the separation of wind turbine noise from background noise in order to assess the conformity of a wind park with respect to the Italian legislation [20]. The procedure developed makes, however, strong assumptions about background noise, namely that residual sound only depends on wind and that wind on the ground is not correlated to wind at hub height. This hypothesis has been reported to work in rural areas [50] but it will not be satisfied everywhere. Furthermore the method may not generalize easily to other community or industrial noise sources.

The receiver perspective seems more relevant when it comes to evaluating community response. However it makes the estimation of specific sounds operator-dependent. With standardization of field noise measurement procedures in mind, this is problematic because it would compromise measurement reproducibility. First, although the average capacity of a healthy human hearing is well documented [51], the dispersion around the average is still to be assessed [52]. Second, hearing loss is a pathology that can go unnoticed for a long time especially when hearing loss does not affect speech comprehension [53]. Moreover hearing tests are not a routine practice of occupational health check-ups in many countries. Third, when listening for specific sound, higher level capacities of the human operator may interfere like attention, concentration and the knowledge of the variability of the source of specific sound.

The receiver perspective also sets limits to the duration of the interval of observation. In the absence of machine-based estimation of specific sound, this task can only be performed over short term measurements, i.e., not beyond a few hours of listening while arguably the compliance of a noisy facility should in general be assessed over a longer time frame for the sake of representativeness. As the variability of the source under investigation and the range between source and receiver increases, so does the uncertainty of the estimation of specific sound.

### 5.3. Residual Sound in Practice

While residual sound is easily told from the total sound in a picture like the one of Figure 1, in reality it is not always straightforward to estimate $L_{res}$, because what is captured by the microphone is the total sound. The source is generally not under the control of the operator carrying the noise impact assessment.

Moreover, in several cases the source operates permanently with a quite stable sound emission. Stopping such a source can be either costly or simply not an option. The power network provides several examples where the estimation of residual sound is challenging. First, turning off a wind farm is possible but results in large losses of revenue for the operator. Second, stopping a large installation like a nuclear power plant is a very long process that raises the issue of the redundancy of the power network so that the loss of production can be compensated by another plant. At the other end of the power networks evaluating the sound emergence of power transformer means a power outage for hundreds of people or strategic services like a hospital.

As already mentioned for the specific sound, blind source separation is not currently available off-the-shelf. In addition, it will be detailed later, modeling residual sound as a substitute for measurements is a difficult task with many unknowns and large uncertainties that are not compatible with low emergence-based limit values. To solve this issue, a common approach is to estimate the residual sound pressure level at another location far enough from the source of specific sound that cannot be stopped. This is of course problematic for both reproducibility and representativeness. Alternatively, it may be possible to benefit from maintenance phases to measure the residual sound pressure level at the right location but then facing the risk of non-contemporaneousness between the estimation of the specific source and the one of the residual sound. The puzzle of the unstoppable source has no perfect solution and all noise limits that refer to $L_{res}$ face this issue, not only sound emergence.

### 5.4. Incoherent Source Assumption

The meaning of “increase” in the definition of sound emergence can also be a matter of interpretation. Currently, “increase” can be understood in the algebraic sense, so that *e* can take negative values. Under the assumption of coherent sources, assuming that the residual sound contains sound from a sound source *A* it is theoretically possible to observe a local decrease in sound pressure level when a sound source *B* is turned on, especially if *A* and *B* produce tonal noise at the same frequency. This may happen in practice for instance in the case of factories that combine several identical units that generate low frequencies.

Destructive interferences, however, are not expected for most of real world noise sources at the usual receiver distances. Indeed, the implicit assumption behind sound emergence is the one of incoherent summation of contributions from residual sound and specific sound so that e≥0 dB.

### 5.5. Variety of Metrics

As mentioned, the ISO 1996-1 standard does not go into details regarding the practical evaluation of sound emergence and especially what metric to use for the quantification of the sound pressure level of specific sound. This aspect is at least partially addressed in a national standard [6]. This French standard leaves the measurement operator free to choose a suitable measure between equivalent sound pressure levels $L_{eq,T}$, where time constant *T* is not specified and fractile sound pressure, levels $L_{X}$, where the threshold is not specified either, derived from $L_{eq}$ time series. The wide freedom left regarding the specification of the metric used for the residual sound and for the total sound may reflect that there is no consensus on suitable metrics for the evaluation of different environmental sound mixtures. The consequence is that different noise consultants may decide to choose different metrics for the same source which can not but compromise the reproducibility of emergence measurements.

## 6. Emergence and Development Planning

Since emergence is defined with respect to residual sound, setting sound-emergence-based noise limits is problematic for all the stakeholders because they are both difficult to predict and difficult to ascertain in the long run.

### 6.1. Seen from the Source

First, emergence is problematic for the owner or the operator of a noisy facility, as emergence-based noise limits offer little assurance about the long-term compliance of the facility to the noise limits. One can assume that the operator has a good command on the noise emissions of the facility and that these emissions can be kept at the same sound pressure level as specified or observed during the environmental impact assessment.

But, conversely, the operator has little control over variations in background noise. For instance a plant *A* could be located close to another noisy plant *B*. In the environmental impact study for plant *A* the noise from plant *B* contributes to the definition of the residual sound in the neighborhood of plants *A* and *B*. If plant *B* goes bankrupt or is relocated, the decrease in residual sound level is to be expected and the compliance of plant *A* with respect to noise is at stake. Other changes in the environment could have similar consequences like the construction of a building or a barrier between plant *B* and the community. This would obviously reduce the contribution of plant *B* to the residual sound level and may compromise the conformity of plant *A*.

Since it is very difficult to predict sound emergence using software simulation tools, emergence will be measured. But the duration of the measurement campaign will be constrained by financial and practical considerations. Therefore, the measurements will typically not be carried out over more than a few days. This prevents the documentation of seasonal variations. If the environmental impact assessment is done well into the vegetation period, then background noise may be strongly influenced by the interaction of wind with foliage while this component will be more or less absent outside the vegetation period in the case of deciduous trees. But vegetation is not the only parameter subject to seasonal variations inducing seasonal variations in residual sound levels. One can also mention seasonal winds, snow cover and seasonal human activities.

### 6.2. The Community Perspective

Second, emergence is problematic for the community because the reference to residual sound in the noise limit sets a shifting baseline [54]. In other words, it offers little protection, if any, against higher noise levels. Let us imagine a pristine rural environment without any noisy activity. Plant *A* may be allowed to operate continuously after the environmental impact assessment because the immission level in the community is not higher than $L_{res}+k$ decibels. If a few years later another factory *C* wants to operate in the very same area, the reference residual sound level for factory *C* will then be $L_{res}+k$, everything else remaining equal, and the consequence of the continuous operation of factory *C* may lead to sound levels as high as $L_{res}+2k$. As new economic developments appear in the area the sound can still increase due to the shifting baseline of the residual sound level. Unless cumulative noise limits are introduced like the one defined in [25] or in the Italian legislation [21], the limit will only be set by shortage of available land.

### 6.3. Predicting Emergence

Third, emergence is problematic because it is very difficult to predict with good confidence. Again, the main issue is the reference to residual sound. Residual sound is a priori a mixture of a wide variety of sources. This raises several challenges. The first is to identify the sources. They may include streams, foliage noise from trees and shrubs, birds during the dawn chorus, elongated structures singing in the wind, diverse appliances and equipments present in the environment—like heat pumps—road and rail traffic and industrial noise. This implies a survey over a wide area.

Provided that this survey be successful, the physical modeling of the emission, the spatial distribution, the duty pattern or cycle of these multiple sources is not an obvious task and requires a lot of effort. Some of the sources listed may seem of minor importance and it is certainly possible to perform a ranking and focus on the most contributing sources but in rural settings, like in the case of wind energy developments, it is likely that the simulation of the residual sound level would require the consideration of sources, the modeling of which is not well established. Moreover, at the time of writing there is no such thing as a reliable macroscopic model for $L_{res}$. The general spectral trends are well known [55] but the calibration is problematic and the evaluation of emergence must always be carried at a specific place with specific sources.

Furthermore, an additional difficulty is brought by the fact that $L_{res}$ may be as low as 30–35 dB. It is well known that the current engineering-level noise prediction methods are not designed for the simulation of such low levels. All this is in stark contrast with the general approach in noise impact studies where limit values are expressed as sound levels. Simulations are a routine operation and if the regulations set limits on the contribution of the source under study, the modeling can focus on the infrastructure/plant/source under investigation and forget the other sources which is a much more feasible task. This allows to investigate yearly-averaged values by considering the seasonal variations of the emissions and of the propagation medium and also to consider long term trends. All of this is beneficial to the stability of the compliance of noisy installations.

## 7. Conclusions and Recommendations

We reviewed the use of sound emergence limit values in legislation and other official documents. Noise limit values referring to background sound appear to be in use in a small number of legal texts throughout the world. The background sound level can be used to define the initial state of the soundscape in a place before setting the maximum allowable contribution for the specific sound source to be regulated or for setting the allowable sound pressure level of the total sound. Some countries, however, prefer to express the noise limit in real time by a direct reference to the difference between total sound pressure level and residual sound pressure level. Occasionally the allowed difference can be as low as 3 dB(A) and is supposed to be obtained from measurements.

In this paper the relevance of sound emergence was assessed from the point of view of perception and annoyance, the one of measurement practice and the one development planning. The literature indicates that there is little evidence, if any, that sound emergence is a better rating than sound pressure level because the background sound is a second order parameter in the determination of annoyance. Moreover, sound emergence does not provide reliable information about the audibility of the specific sound. From a practical perspective the measurement of emergence raises several issues relating to the understanding of the specific sound, the access to the background sound and measurement uncertainty. In addition, sound emergence is problematic from a planning perspective because this indicator relies on the shifting baseline of residual sound. Therefore sound emergence offers little guarantee of compliance to the different stakeholders. Furthermore, the necessity to predict background sound makes the simulation of emergence in software more challenging and uncertain than the one of the sound pressure level from a specific source.

From all this it would seem reasonable to reconsider the use of sound emergence in legislations that rely on it. The weight of history is not enough and research is needed to provide evidence that sound emergence is relevant for setting environmental noise limits. Further research about the potential correlation between annoyance and sound emergence from specific sources is necessary. Due to the rapid and global development of wind energy, the justification of the use of sound emergence in the environmental impact assessment of wind farms is certainly worth investigating. Since sound emergence appears to be a poor predictor of audibility, the temporal and spectral characteristics of the specific source should be taken into account in the estimation of sound emergence leading to source-specific choices of the metrics used for estimating total and residual sound pressure levels.

If switching to another metric is out of question we can make the following recommendations: (1) the specific sound should be defined from the source perspective, (2) the metrics used for residual and specific sound should be specified unambiguously, (3) the estimation of emergence should be based on long term measurements to account for the variability of the residual and the specific sound and (4) sound emergence should be used in combination with limits based on sound levels to avoid the shifting baseline phenomenon. Attempts have been already made in this direction but they are only incompletely successful.

In our opinion and in the wake of [35], combining (i) an estimation of the audibility of the specific sound for the community and (ii) an estimation of $L_{spec}$ would be by far superior to sound emergence while serving the same purposes. The audibility assessment could be built on the above-mentioned previous research [41]. Regarding $L_{spec}$, this quantity should be obtained with a sufficient signal-to-noise ratio. This may require the acceptance that source-attributable noise at the receiver may not always be accessible to direct measurements of sound pressure level and that simulations may be necessary.

## Figures and Tables

**Figure 1 ijerph-16-04517-f001:**
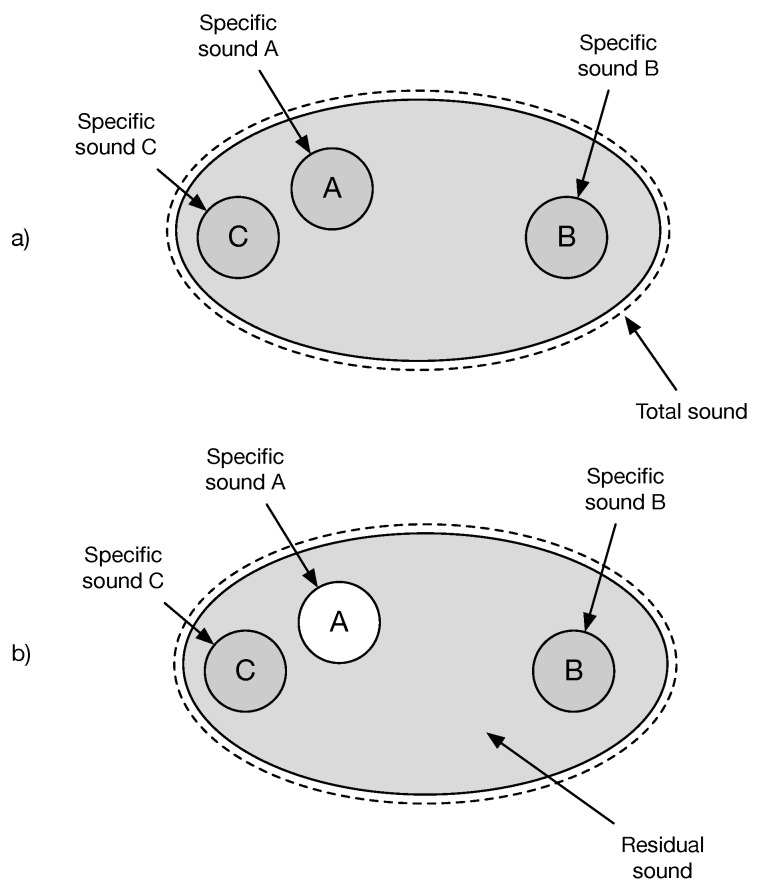
Total, specific and residual sound. 3 sources A, B and C are identified but other non identified sources combine into the total sound (**a**). With respect to A, the residual sound is observed when the specific sound A is absent (**b**), everything else being equal.

**Figure 2 ijerph-16-04517-f002:**
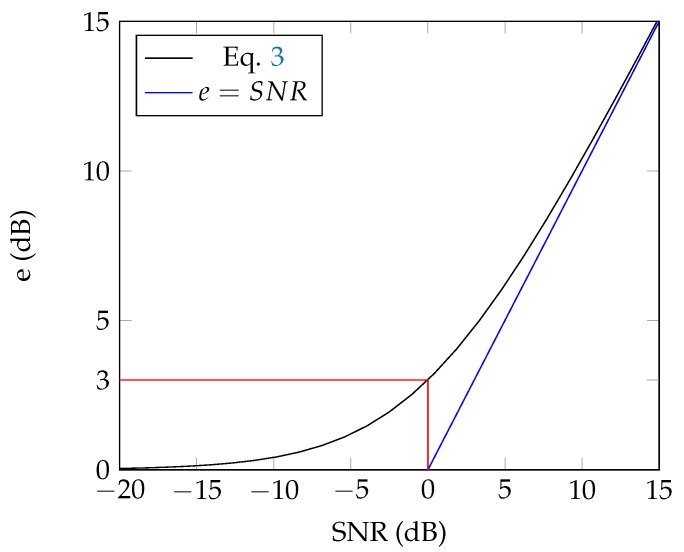
Relationship between signal-to-noise ratio (SNR) and *e*.

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
