# Peer review of "Challenges of the Use of Sound Emergence for Setting Legal Noise Limits"

_ijerph, 2019, doi:10.3390/ijerph16224517_

Round 1

Reviewer 1 Report

This is an excellent paper, describing the main challenges of the use of emergence-based metrics when defining legal noise limits.  The manuscript is very well-written and refers to multiple examples of current practice in legislation in several countries.  

There are only a couple of typos (see e.g. line 123), and levels in line 150 should be further defined in the context of BS4142.

The only issue I have with the paper (in its current state) is that the challenges are described in a very comprehensive and clear manner, but there is a limited description and discussion on the recommendations.  I encourage the authors to expand that section, referring to specific cases, as done for the challenges.

Author Response

There are only a couple of typos (see e.g. line 123) : 

We corrected a few typos including this one.

and levels in line 150 should be further defined in the context of BS4142.

The existing sentence :

« For these sources the base indicators to be used is LAeqT for both the specific and the background noise »

was expanded

« The noise indicators specified in \cite{BSI:2014aa} are $L_{Aeq}$ for $L_{tot}$, $L_{spec}$ and $L_{res}$, and $L_{AF90}$ for background noise. The background noise is not identical to the residual sound (Cf section \ref{sec:def}) since background noise is estimated when the source under investigation is operating whereas $L_{spec}$ assumes that the source under investigation is turned off.»

The whole paragraph was reworked to make it consistent with the definitions of section 2.

The only issue I have with the paper (in its current state) is that the challenges are described in a very comprehensive and clear manner, but there is a limited description and discussion on the recommendations. I encourage the authors to expand that section, referring to specific cases, as done for the challenges.

The aim of the paper is not to recommend the use of sound emergence but to encourage further research regarding the correlation of sound emergence with annoyance and this research should be source-specific. Moreover, we emphasize that sound emergence is more difficult to measure and to predict than other metrics. Expanding the recommendations would require to be source-specific - while the paper deals with sound emergence - and warrants further research leading to specific papers.

The second last paragraph of the paper was rearranged and slightly expanded regarding the need for further research (new text in yellow).

From all this it would seem reasonable to reconsider the use of sound emergence in legislations that rely on it. The weight of history is not enough and research is needed to provide evidence that sound emergence is relevant for setting environmental noise limits. Further research about the potential correlation between annoyance and sound emergence from specific sources is necessary. Due to the rapid and global development of wind energy, the justification of the use of sound emergence in the environmental impact assessment of wind farms is certainly worth investigating. Since sound emergence appears to be a poor predictor of audibility, the temporal and spectral characteristics of the specific source should be taken into account in the estimation of sound emergence leading to source-specific choices of the metrics used for estimating total and residual sound pressure levels, If switching to another metric is out of question we can make the following recommendations: (1) the specific sound should be defined from the source perspective, (2) the metrics used for residual and specific sound should be specified unambiguously, (3) the estimation of emergence should be based on long term measurements to account for the variability of the residual and the specific sound, (4) sound emergence should be used in combination with limits based on sound levels to avoid the shifting baseline phenomenon. Attempts have been already made in this direction but they are only incompletely successful.

Reviewer 2 Report

The paper is brilliant, really clear and perfectly presented and discussed. The whole demonstration is precise and easy to follow.

The paper starts with a complete description of the emergence notion and the authors explain it and considers the different ways it can be done (in regards with Member state regulation) i-e from by a difference of intensity of sound (dB oand dB(A) and/or by a difference in frequencies.

Residual sound also is properly discussed and actually well defined in the problematic part of the paper.

Many EE regulations (+ Australian) are compared and despite of the lack of few official texts, authors manage to write an interesting comparative analysis.

Paragraph 4 is maybe the weakest paragraph because we believe that annoyance and perception cannot be explained only by the acoustics metrics. Of course, many studies have shown some links between dB (or frequential) emergence and annoyance (all the studies of CSTB) but regression curves found are still not able to describe the phenomenon. Actually, authors might have refered some works made by sociologists that tries to show that annoyance are not only explained by acoustic measurements. Perception is an action (Berthoz Alain) and we cannot measure the action of the user that will focus on a specific emergence and associate a noise with annoyance. Of course, the signal has some properties (emergences), but we think that there is not “evident” link between a degree of annoyance and some sound properties.

It is for this reason that we think that the authors are aware of this point and thus launch a series of criticisms on the different metric possibilities of the notion of emergence. I quote the authors.

Sound emergence is a second order descriptor of annoyance Emergence is only a proxy for SNR when SNR is high Sound emergence does not reflect detectability The relationship between annoyance and the strength of sound emergence

This criticism is pursued with regard to the measurement and especially the need for the presence of an operator during the measurement to document the source potentially awkward emergence. I quote : “line264-265 : It will rather require the presence of a human operator who will be able to detect the specific sound in the total sound”.  It is actually what I learnt to do when I was working for the noise service of a Town Hall in France and it’s actually exactly how was built the software we used (01dB- source codage).

Once again the authors point out, this time with regard to measurement, the difficulty of associating an emergent source with the expression of an annoyance. This part is perfectly discussed in our opinion.

The paper continues its reflection with the theme of Emergence and planning. Authors explain with logic the limits to use emergence criteria in order to set a limit values for communities’ exposition when a place is always evolving and background noise is always increasing by welcoming news activities. Discussion is perfectly logic and again strong criticism on emergence strategies are raised up.

In conclusions, perfect resumé is proposed and it opens up to very interesting debate on the use of such metrics.

one detail
line 224 : While it has been illustrated in Figure ?? .. 2 questions marks instead of figure 2 ?

Author Response

Paragraph 4 is maybe the weakest paragraph because we believe that annoyance and perception cannot be explained only by the acoustics metrics. Of course, many studies have shown some links between dB (or frequential) emergence and annoyance (all the studies of CSTB) but regression curves found are still not able to describe the phenomenon. Actually, authors might have refered some works made by sociologists that tries to show that annoyance are not only explained by acoustic measurements. Perception is an action (Berthoz Alain) and we cannot measure the action of the user that will focus on a specific emergence and associate a noise with annoyance. Of course, the signal has some properties (emergences), but we think that there is not “evident” link between a degree of annoyance and some sound properties.

The authors totally agree with the fact that noise only explains part of the variance in annoyance evaluations. This is already suggested in our account of (Fields 1993) :"Nonetheless transportation noise encompasses a wide variety of sounds, both steady and unsteady, with or without tonalities and the various levels of ownership among the respondents with respect to the source of specific sound."

Still it makes sense to search for noise metrics that correlate reasonably well with annoyance.

We tried to make this aspect clearer in our paper. The very first sentences of "Emergence, annoyance and perception":

While one can expect that the metrics used for setting noise limit values be connected with health effects in the broad sense, and annoyance in particular, very little research was done on the merits of sound emergence from a public health perspective.

are replaced by 

It is well known that noise annoyance is only partly determined by acoustic factors

(Fields,1993 ;Stallen ,1999). One can expect, however, that the metrics used for setting noise limit values be connected with health effects in the broad sense, and annoyance in particular, It appears that very little research was done on the merits of sound emergence from a public health perspective.

line 224 : While it has been illustrated in Figure ?? .. 2 questions marks instead of figure 2 ?

This was corrected and a few other typos as well.